Updated range distribution of the non-native Asian green mussel Perna viridis (Linnaeus, 1758) at Guanabara Bay, Rio de Janeiro, Brazil

de Messano Luciana V. R. lvicentebm@gmail.com
Gonçalves José E. A.
Kassuga Alexandre D.
http://orcid.org/0000-0001-8921-8767 da Silva Alexandre R.
Masi Bruno P. masibruno@gmail.com
Messano Héctor F.
Fardin Denny
Coutinho Ricardo
Marine Biotechnology Department, Instituto de Estudos do Mar Almirante Paulo Moreira , Arraial do Cabo, Rio de Janeiro , Brazil
Sunny Armando
Electronic publication date: 2024 Dec 19
Publication date: 2024
Volume: 12
Electronic Location ID: e18649
Received 2024 Aug 6; Accepted 2024 Nov 15
Copyright: © 2024 de Messano et al.
Copyright year: 2024
Copyright holder: de Messano et al.
License: This is an open access article distributed under the terms of the Creative Commons Attribution License, which permits unrestricted use, distribution, reproduction and adaptation in any medium and for any purpose provided that it is properly attributed. For attribution, the original author(s), title, publication source (PeerJ) and either DOI or URL of the article must be cited.
License URL: https://creativecommons.org/licenses/by/4.0/

Keywords: Marine invasive species, Expansion, Perna viridis, South Atlantic

Funding: Cooperation agreement between IEAPM and PETROBRAS (ANP N° 22003-8 regulated by R, D&I investment clauses of Brazilian Agency of Petroleum, Natural Gas and Biofuels - ANP Resolution 03/2015) This work was supported by a cooperation agreement between IEAPM and PETROBRAS (ANP N° 22003-8 regulated by R,D&I investment clauses of Brazilian Agency of Petroleum, Natural Gas and Biofuels - ANP Resolution 03/2015). The funders had no role in study design, data collection and analysis, decision to publish, or preparation of the manuscript.

==============================
Guanabara Bay, located at Rio de Janeiro, Brazil, is a highly urbanized and polluted estuary that houses different port areas, shipyards, and marinas of intense maritime traffic. This infrastructure is widely associated with the introduction and spread of non-native sessile species. A rapid assessment of non-native benthic sessile species conducted in the bay in late 2022 across 19 sites identified a total of 83 taxa, both native and non-native, classified into the following main groups: one Cyanophyta, 13 Macroalgae, 14 Porifera, 11 Cnidaria, six Bryozoa, five Annelida, 10 Mollusca, six Crustacea, 10 Echinodermata, and seven Ascidiacea. Our findings revealed the proliferation of the Asian green mussel (Perna viridis Linnaeus, 1758), a species noted for its exceptional ability to achieve extremely high biomass levels globally. In Brazil, the bivalve was first reported less than 6 years ago in 2018 at Guanabara Bay, on a mariculture farm at Arraial do Cabo (200 km away) in 2023 and more recently in the south (Paranaguá Bay), besides two coastal islands outside Guanabara Bay on natural rocky shores. The present survey recorded P. viridis at 17 sites, including natural substrata, co-occurring with native species. No Tubastraea spp. were observed in Guanabara Bay. Controlling and mitigating the consequences of bioinvasion events can be challenging, but biosafety protocols should be adopted in the near feature to minimize the risks and impacts caused by species dispersal.

Introduction

The processes of biological invasion are a global concern, particularly in marine environments, as they can significantly impact coastal marine ecosystems through the introduction of non-native species (NNS) (Pyšek et al., 2020). Marine bioinvasions are associated with multiple pathways but are strongly related to maritime activities, with ballast water discharge and biofouling on vessels serving as major vectors for species transference (Williams et al., 2013). Further, urbanized environments provide novel habitats that can support the establishment and proliferation of NNS post-arrival, acting as stepping stones to natural ecosystems (Dafforn, 2017). The establishment of NNS in a new habitat can disrupt local biodiversity, alter ecosystem functioning, and impose economic costs, representing a direct threat to native marine communities in coastal ecosystems (Bax et al., 2003). Coastal areas, such as estuarine environments, are particularly vulnerable to invasion not only due to the natural dynamics of these ecosystems but also because of the impacts of anthropogenic activities and the urbanized infrastructure historically added to the coastal zones (Johnston et al., 2017).

Guanabara Bay is a fully urbanized estuarine system located in Rio de Janeiro, Brazil, with significant socioeconomic, strategic, sanitary, environmental, and ecological importance—the bay is surrounded by rocky shores and islands (Fistarol et al., 2015). As the subject of numerous research studies, the bay is a well-studied ecosystem stressed by several types of anthropogenic impacts, such as chemical pollution, sewage discharge, heavy metals contamination, floating trash, as well as aquaculture structures and maritime public transportation (Amador, 1997; Coelho, 2007; Fistarol et al., 2015; Soares-Gomes et al., 2016; Fries et al., 2019). The bay hosts the second longest bridge in Latin America—a 13.3 km box girder bridge connecting the cities of Rio de Janeiro and Niterói comprising 103 submersed pillars (Alencar, 2016). Additionally, the bay hosts the Port of Rio de Janeiro (the third main port of Brazil) which supports intense domestic and international maritime traffic, shipyards, marinas, and fisheries facilities (ANTAQ, 2023). This type of infrastructure is widely associated with the introduction and spread of non-native sessile species and thus bioinvasion events have been reported as one of the main threats to the bay (Soares-Gomes et al., 2016). The first NNS record was Styela plicata (Lesueur, 1823) in the 19th century (Simões, 1981) but recent reports continue to document new occurrences. As the bay is surrounded by natural consolidated substrates and also operates as an important maritime hub, it plays an important role as both a donor and recipient area for the main vectors, namely ballast water, biofouling, and also marine debris (Therriault et al., 2021).

Less than 6 years ago, de Messano et al. (2019) first recorded the presence of Perna viridis (Linnaeus, 1758) in the South Atlantic on experimental plates placed at Guanabara Bay. At that time, researchers found seven individuals of P. viridis: two individuals on plates, two on pilings, and three on a seawall. The Asian green mussel is a marine bivalve native to the western Indo-Pacific region, and it is considered a high-risk invader and a threat worldwide (Dias et al., 2018). Since this first record, new introductions on artificial substrates have been reported in Brazilian waters: at Arraial do Cabo, in the state of Rio de Janeiro (dos Santos, Bertollo & Creed, 2023) and recently in the state of Paraná (Beltrão et al., 2024). In June 2023, Machado et al. (2023) reported the first record of the Asian green mussel on natural rocky shores in two coastal islands outside Guanabara Bay (Tijucas Islands). The conspicuous spread of invasive species, such as P. viridis, increase environmental concerns and highlights the importance of prevention and surveillance actions.

Given this scenario and the significant importance of this ecosystem, we conducted a rapid assessment to investigate sessile species distribution around the bay and to survey non-native species. To estimate the spread of P. viridis towards other areas in Brazil, we used Ecological Niche Modelling (ENM), an important tool to calculate the potential dispersion of non-indigenous species (Bumbeer et al., 2018; Li et al., 2019; da Conceição et al., 2023; Trevisan et al., 2023).

Materials and Methods

Study site

A total of 19 sites were strategically chosen in Guanabara Bay to survey sessile species and to verify the occurrence of non-native species across the bay. Guanabara Bay is a semi-enclosed coastal environment, connected to the sea by a 1.6 km-wide narrow access channel, exhibiting hydrographic heterogeneity across its various areas. Its water quality exhibits non-uniform characteristics: salinity tends to be higher near the bay’s mouth, whereas pollution levels increase toward the bay’s interior (Fries et al., 2019). In their study, Paranhos et al. (2001) reported mean total nitrogen values ranging from 0.6 to 68.3 µM near the bay’s entrance and from 5 to 346 µM in its center. Additionally, this revealed mean total phosphorus values of 0.05 to 7.4 µM near the bay’s entrance and 0.2 to 26.4 µM in the center.

Biological data

Between September and October 2022, scuba dives were conducted to document sessile species and survey non-native species on natural substrates (consisting of rocky shores), as well as artificial substrates such as granite boulder breakwaters, piers, and concrete pilings, located at the entrance and inner parts of Guanabara Bay, Rio de Janeiro, Brazil.

Due to low visibility, areas of 100 cm2 were photographed at each site using a Nikon D7000 DSLR camera with a Nikkor AF 60 mm (macro) lens, enclosed in a DXD7000 underwater housing (Sea & Sea), and illuminated artificially with two YS110 flashes (Sea & Sea). Three images were captured every 5 m along 100 m horizontal transects positioned at 4 and 8 m depending on the depth of the site. A different approach was used for the President Costa e Silva bridge (Rio-Niterói) pillars due to their distinct shape. Two divers swam around the selected pillars at two different depths to photograph the organisms.

Photographs were analyzed using the software Coral Point Count with Excel Extensions CPCe 4.1 (Kohler & Gill, 2006) as a tool to enhance the images and support the qualitative analysis. The taxa in each image were identified at the lowest taxonomic level possible using morphological characterization, dichotomous keys and/or consulting experts. Data were converted into a presence/absence matrix per site and at the respective depths (when applicable).

Ecological niche modelling

Perna viridis records for ecological niche modelling were obtained from the Global Biodiversity Information Facility (GBIF, https://www.gbif.org). First, we filtered the occurrence data to eliminate duplicates and records with incorrect data. Subsequently, we selected the records from its native range based on Rajagopal et al. (2006) and dos Santos, Bertollo & Creed (2023), totaling 190 records (Table S1). Data on environmental predictors were obtained from Bio-Oracle (https://www.bio-oracle.org), a global environmental dataset designed for marine species distribution modeling with a resolution of 5 arc min (Tyberghein et al., 2012; Assis et al., 2018, 2024). A total of 14 layers were downloaded for the following variables: mean depth, mean dissolved iron, mean nitrate, mean dissolved oxygen, mean pH, mean primary productivity, mean phosphate, mean silicate, mean dissolved chlorophyl, range and mean salinity, range and mean temperature, and mean current velocity. The environmental layers were cropped into polygons representing the biogeographical provinces (Spalding et al., 2007) of the native range of P. viridis. To avoid autocorrelation between variables, we estimated a Pearson’s correlation between all possible combinations. Correlated variables (r ≥ 0.7) were selected based on the species ecology of P. viridis, of which six—salinity range, mean primary productivity, mean dissolved oxygen, mean silicate, temperature range and mean current velocity—were chosen to run the modelling analysis.

Perna viridis ecological niche was modeled using the maximum entropy routine implemented in Maxent version 3.3.3 (Phillips, Anderson & Schapire, 2006; Elith et al., 2011). The model was calibrated using 30% for testing and 50 replicates were performed using cross validation. Model accuracy was evaluated using Area Under the Curve (AUC) and True Skill Statistics (TSS), two metrics important for evaluating ecological niche models (Fielding & Bell, 1997; Allouche, Tsoar & Kadmon, 2006). Data from the average model was then used to predict the environmental suitability area for P. viridis in the Brazilian regions where the species has been recorded. All analysis were conducted using the SDM package (Naimi & Araújo, 2016) built for R (R Core Team, 2024).

Results

The benthic assemblage survey of 19 natural and artificial substrates identified 83 taxa. They were classified into major groups that included one Cyanophyta, 13 macroalgae, 14 Porifera, 11 Cnidaria, six Bryozoa, five Annelida, 10 Mollusca, six Crustacea, 10 Echinodermata and seven Ascidiacea (Table S2). Among them, we registered 15 non-native species, namely: the sponge Paraleucilla magna Klautau, Monteiro & Borojevic, 2004; the polychaete Branchiomma luctuosum (Grube, 1870); the bivalves Isognomon bicolor (Adams, 1845), Saccostrea cuccullata (Born, 1778), Perna perna (Linnaeus, 1758) and Perna viridis (Linnaeus, 1758); the barnacles Megabalanus coccopoma (Darwin, 1854), Amphibalanus amphitrite (Darwin, 1854) and Balanus trigonus Darwin, 1854; the bryozoans Bugula neritina (Linnaeus, 1758) and Schizoporella errata (Waters, 1878); the ascidians Styela plicata (Lesueur, 1823), Clavelina oblonga Herdman, 1880 and Didemnum perlucidum Monniot F., 1983, and the ophiuroidea Ophiothela mirabilis (Verrill, 1867). These species were already recorded at Guanabara Bay and other parts of the Brazillian coast (Table 1).

Table 1 List of non-native species identified during the assessment survey.

The type of substrate on which the organisms were observed are natural, artificial or both.

Taxa	Substrate	Origin	References to Guanabara Bay occurrences	
PORIFERA: CALCAREA				
Paraleucilla magna Klautau, Monteiro & Borojevic, 2004	Both	Mediterranean Sea	Batista et al. (2013)	
ANNELIDA: POLYCHAETA				
Branchiomma luctuosum (Grube, 1870)	Both	Red Sea	Oricchio et al. (2019)	
MOLLUSCA: BIVALVIA				
Isognomon bicolor (Adams, 1845)	Both	Caribbean Sea	Puga et al. (2019)	
Saccostrea cuccullata (Born, 1778)	Both	Indo-West Pacific	Puga et al. (2019)	
Perna perna (Linnaeus, 1758)	Both	Red Sea, Eastern and Southwestern Africa	Oricchio et al. (2019)	
Perna viridis (Linnaeus, 1758)	Both	Eastern and Western Indian Ocean	de Messano et al. (2019)	
CRUSTACEA: CIRRIPEDIA				
Amphibalanus amphitrite (Darwin, 1854)	Both	Unknown	Puga et al. (2019)	
Balanus trigonus Darwin, 1854	Both	Unknown	Oricchio et al. (2019)	
Megabalanus coccopoma (Darwin, 1854)	Both	Unknown	Oricchio et al. (2019)	
BRYOZOA				
Bugula neritina (Linnaeus, 1758)	Both	North East Pacific	Puga et al. (2019)	
Schizoporella errata (Waters, 1878)	Both	Mediterranean Sea	Soares-Gomes et al. (2016)	
TUNICATA: ASCIDIACEA				
Clavelina oblonga Herdman, 1880	Both	Bermudas	Simões (1981)	
Didemnum perlucidum Monniot, 1983	Both	Caribbean Sea	Oricchio et al. (2019)	
Styela plicata (Lesueur, 1823)	Artificial	West Pacific	Oricchio et al. (2019)	
ECHINODERMATA: OFIUROIDEA				
Ophiothela mirabilis (Verril, 1897)	Both	Eastern Pacific	Machado et al. (2023)	

We photographed 73 different P. viridis individuals co-occurring with the local community (Fig. 1A). The bivalve was registered in different sizes, occupying 17 sites around the bay (on both artificial and natural substrates) (Fig. 1B) and co-occurring with Perna perna (Fig. 1C). Neither was registered at Pai Island and the co-occurrence with Perna perna was not observed in the Itaipu rocky shores, where only P. perna was recorded.

Figure 1 Perna viridis photographs taken during the survey.

(A) Co-occurring with other fouling species. (B) Different size of Perna viridis. (C) With the co-generic Perna perna.

The average model presented an AUC of 0.94 and a TSS of 0.78 indicating a good fit. Based on AUC, the relative importance of environmental variables was as follows: salinity range (48.5%), silicate mean (5.8%), primary productivity mean (2.3%), temperature range (1.0%), dissolved oxygen mean (0.6%) and velocity range (0.3%). As for environmental suitability, the predictive model indicated high suitability for the states of Paraná and Rio de Janeiro (Fig. 2A). Inside Guanabara Bay, the environmental suitability ranged from 50% to 70%; in the surrounding areas it ranged from 30% to 70%, especially around ports. The environmental suitability map for P. viridis in its native area is available in the (Fig. S3).

Figure 2 Habitat suitability maps of the non-native Asian green mussel Perna viridis and its occurrence.

(A) Habitat suitability for the coast of Brazil and dots indicates literature records of P. viridis. (B) Habitat suitability for the coast of Rio de Janeiro. (C) Map showing Guanabara Bay location at Rio de Janeiro state, Brazil. 1. Ilha do Pai [−43.092, −22.98571]; 2. Ponta de Itaipu [−43.056054, −22.9735]; 3. Ilha de Cotunduba [−43.143096, −22.96347]; 4. Ilha do Veado [−43.10799, −22.958]; 5. Costão Praia da Urca [−43.16168, −22.937]; 6. Marina da Gloria [−43.162, −22.922]; 7. Ilha de Boa Viagem [−43.12940, −22.9145]; 8. UFF Gragoatá [−43.1399, −22.89839]; 9. Ilha da feiticeira [−43.169, −22.885]; 10. Pilar 108: Ponte Rio-Niterói [−43.145, −22.878]; 11. Pilar 111: Ponte Rio-Niterói [−43.14546, −22.866]; 12. Pilar 67: Ponte Rio-Niterói [−43.18292, −22.8735]; 13. Pilar 66. Ponte Rio-Niterói [−43.182, −22.863]; 14. Ilha De Santa Cruz [−43.11571, −22.85397]; 15. Pedras da passagem [−43.15999,−22.84477]; 16. Ilha do Engenho [−43.122, −22.83297]; 17. Ilha das Palmas [−43.15553, −22.79467]; 18. Rochedo da Praia dos Frades [−43.1105, −22.774]; 19. Praia Dos Frades [−43.098, −22.772].

Discussion

Perna viridis is a recent non-native species recorded in South Atlantic waters (de Messano et al., 2019). It is native to Southeast Asia (Siddall, 1980), with additional studies identifying its native range as including Hong Kong, Taiwan, and southwest China (Baker et al., 2007; Dias et al., 2018). The species was first recorded in the Atlantic basin in the 1990s at the Caribbean Sea and currently its population distribution is sparse, occurring at discrete points (Gobin et al., 2013). First detected at the Trinidad Island in 1990, it was then reported in Venezuela in 1993, first at the northern state of Sucre and later at La Restinga Lagoon, in Isla Margarita (Bigatti, Miloslavich & Penchaszadeh, 2005), and later, in 1998, in Kingston Harbor (Buddo, Steele & D’Oyen, 2003). Perna viridis was recorded in 1999 at Tampa Bay (Gulf of Mexico) and it spread into two more locations on the Florida coast (USA), where the established populations are declining (Firth, Knights & Bell, 2011; Galimany et al., 2018; Levine, Granneman & Geiger, 2022). In Brazil, after four previous records of P. viridis on artificial substrates (de Messano et al., 2019; Soares et al., 2022; dos Santos, Bertollo & Creed, 2023; Beltrão et al., 2024). Machado et al. (2023) first identified the species on natural substrates in the South Atlantic.

Events of P. viridis invasion have been related to ship hull fouling and to ballast water discharge (Mead et al., 2011; Minchin et al., 2016). More recently, rafting on marine debris in Colombia was proposed as a secondary dispersion mechanism (Gracia & Rangel-Buitrago, 2020). At Guanabara Bay, P. viridis has been found attached on a floating Styrofoam (Machado et al., 2023). A key component of bioinvasion dynamics after the first introduction is the establishment of non-native species on natural substrates after a secondary dispersion (Bailey et al., 2020). Perna viridis dispersion in the bay is clearly an on-going process, since it was first detected less than six years ago and is only now established around the bay, attached to artificial and natural substrates and co-occurring with the native community. Predicting which non-native species will become invasive is unlikely (Therriault et al., 2021), but the rapid dispersion observed suggests a high invasive potential in this area. Gilg et al. (2014) suggested that the larvae of P.viridis likely dispersed 10 km in northeast Florida in three years. Based on our observations in Guanabara bay, P. viridis was found approximately 13 km from its first record six years ago.

Among the 19 sites visited, P. viridis co-occurred in eight with the congeneric Perna perna, which can be found on several rocky shores at Guanabara Bay (Ansari et al., 2016). Perna perna is an ancient invader in Brazil (Silva et al., 2018) and the environmental requirements are similar for both species, with both co-occurring worldwide (Micklem et al., 2016). Perna viridis is considered a superior competitor, showing higher thermal and salinity tolerance limits than P. perna, thus highly adapted to polluted areas (Rajagopal et al., 2006). Moreover, P. perna displacement was already observed in Venezuela (de Bravo, Chung & Perez, 1998). Since the species is an important economic resource for the collector communities around the bay (Lage & Jablonski, 2008), an eventual displacement may cause an economic impact. Conversely, studies of P. perna and P. viridis in Venezuela showed that P. perna presents a higher growth rate compared with P. viridis in suspended culture (Urbano et al., 2005; Acosta et al., 2009), whereas P. viridis reached a higher growth rate in bottom culture systems (Acosta-Balbás et al., 2019). At Guanabara Bay, P. perna is reared in longline systems near the bay’s mouth (Lage & Jablonski, 2008).

Our results indicate that the habitat suitability of P. viridis is correlated with relatively eutrophic areas. Increase in chlorophyll-a concentration often leads to greater food availability for primary consumers such the bivalves. These organisms respond by increasing their abundance, biomass, and assimilation efficiency (Moraitis et al., 2018) but it is important to note that these models disregard local factors such as predation, competition, microclimate conditions and climatic refugee, among others. These features also dictate if an area might favor or not the colonization of non-native independent from suitability (Meynard, Leroy & Kaplan, 2019). Nevertheless, areas where the green mussel was recorded, such as the Paranaguá Estuarine Complex (the most recent record), and Guanabara Bay appear to respond positively to organic enrichment and elevated chlorophyll-a levels, corroborating the features used by Rudianto et al. (2024) to identify suitable habitats for P. viridis cultivation. The map reveals that the updated range distribution of P. viridis in Brazil coincides with regions of higher environmental suitability, highlighting the potential of Species Distribution Modeling (SDM) to support environmental management. It is important to note that the model presented in this study indicates low environmental suitability for P. viridis on the Brazilian semiarid coast (Fig. 2). Soares et al. (2022) included P. viridis in a baseline assessment of introduced marine species along this extensive coast. However, the specimens used to confirm the presence of P. viridis in the Northeast region of Brazil were reanalyzed and Arruda et al. (2024) concluded that the individuals collected are not P. viridis, but Mytella strigata (Hanley, 1843), a species native to the South American coast.

Soares-Gomes et al. (2016) reported the non-native species that have successfully established in Guanabara Bay. Most were identified in the present survey and we found no new records (Table 1). It is worth noting that we observed no sun-coral (Tubastraea spp.), even at the sites on the bay’s entrance, despite the record in islands outside Guanabara Bay (Machado et al., 2023). Sun-coral is an important invasive marine species that successfully invaded the eastern Caribbean Sea in the late 1930s, probably introduced by navigation activity. Decades later, in the 1990s, sun coral was first recorded in Brazil (Creed et al., 2017). In the Gulf of Mexico, T. coccinea has been reported attached on oil platforms since the 1970s until 2002, when it was first recorded at the Flower Garden Banks on natural substrates (Fenner & Banks, 2004). Machado et al. (2023) reported the presence of sun-coral in the aforementioned islands, especially in depths > 10 m, in a Marine Protected Area. These results suggest that the environmental conditions of inner Guanabara Bay does not provide conditions for sun-coral colonization and/or survival.

Public and private sector agencies must conduct long-term biodiversity monitoring programs in the area and include bioinvasion studies, since controlling and mitigating the consequences from bioinvasion events can be challenging. To accomplish that, biosecurity protocols should be adopted to minimize the risks and impacts of non-native species (Olenin et al., 2014). Future steps should include multilateral efforts for conducting a deep survey to monitor and verify ecological interactions and impacts. Further, discussion about the impacts and control of non-native species should focus on feasible actions. Brazilian governmental decision-makers are currently developing a framework to prevent bioinvasions and strategic interventions as control actions require urgent implementation.

Conclusions

This study provides crucial insights into the proliferation of the Asian green mussel, Perna viridis, in Guanabara Bay. The application of Species Distribution Modeling (SDM) demonstrates its potential to support environmental management strategies. In the future, long-term biodiversity monitoring programs should be implemented in the area. Furthermore, adopting robust biosecurity protocols is essential to mitigate the risks and impacts associated with non-native species. Notably, among the 15 non-active taxa surveyed, no sun corals (Tubastraea spp.) were observed, even in areas close to the bay entrance.

Supplemental Information

Supplemental Information 1 Records of Perna viridis on its native area used for ecological niche modellings.

Records were download from Global Biodiversity Information Facility https://www.gbif.org/.

Supplemental Information 2 Presence of each identified taxa in each site.

1 - Ilha do Pai [-43.092, -22.98571]; 2 - Ponta de Itaipu [-43.056054, -22.9735]; 3 - Ilha de Cotunduba [-43.143096, -22.96347]; 4 - Ilha do Veado [-43.10799, -22.958]; 5 - Costão Praia da Urca [-43.16168, -22.937]; 6 - Marina da Gloria [-43.162, -22.922]; 7 - Ilha de Boa Viagem [-43.12940, -22.9145]; 8 - UFF Gragoatá [-43.1399, -22.89839]; 9 - Ilha da feiticeira [-43.169, -22.885]; 10 - Pilar 108 - Ponte Rio-Niterói [-43.145, -22.878]; 11 - Pilar 111 - Ponte Rio-Niterói [-43.14546, -22.866],; 12 - Pilar 67 - Ponte Rio-Niterói [-43.18292, -22.8735]; 13 - Pilar 66 - Ponte Rio-Niterói [-43.182, -22.863]; 14 - Ilha De Santa Cruz [-43.11571, -22.85397]; 15 - Pedras da passagem [-43.15999, -22.84477]; 16 - Ilha do Engenho [-43.122, -22.83297]; 17 - Ilha das Palmas [-43.15553, -22.79467]; 18 - Rochedo da Praia dos Frades [-43.1105, -22.774]; 19 - Praia Dos Frades [-43.098, -22.772].

Supplemental Information 3 The environmental suitability map for P. viridis at its native area.

Supplemental Information 4 Occurrence data of the bivalves Perna Perna and Perna Viridis in the Guanabara Bay, Rio de Janeiro, Brazil.

Both species are absent = absent. Presence of only Perna perna = perna. Presence of only Perna viridis = viridis. Both are present = both.

Additional Information and Declarations

Competing Interests

Author Contributions

Data Availability

The authors declare that they have no competing interests.

Luciana V. R. de Messano conceived and designed the experiments, performed the experiments, analyzed the data, prepared figures and/or tables, authored or reviewed drafts of the article, and approved the final draft.

José E. A. Gonçalves conceived and designed the experiments, performed the experiments, analyzed the data, authored or reviewed drafts of the article, and approved the final draft.

Alexandre D. Kassuga conceived and designed the experiments, performed the experiments, analyzed the data, prepared figures and/or tables, authored or reviewed drafts of the article, and approved the final draft.

Alexandre R. da Silva analyzed the data, prepared figures and/or tables, authored or reviewed drafts of the article, and approved the final draft.

Bruno P. Masi analyzed the data, prepared figures and/or tables, authored or reviewed drafts of the article, and approved the final draft.

Héctor F. Messano performed the experiments, authored or reviewed drafts of the article, and approved the final draft.

Denny Fardin performed the experiments, authored or reviewed drafts of the article, and approved the final draft.

Ricardo Coutinho conceived and designed the experiments, analyzed the data, authored or reviewed drafts of the article, and approved the final draft.

The following information was supplied regarding data availability:

The raw data are available in the Supplemental File.

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
