# Peer review of "Updated range distribution of the non-native Asian green mussel Perna viridis (Linnaeus, 1758) at Guanabara Bay, Rio de Janeiro, Brazil"

_PeerJ, doi:10.7717/peerj.18649_

## Round 0.1 · original submission · Minor Revisions

Dear Authors

I am pleased to observe that the revisions made have significantly improved the manuscript. However, the reviewers have suggested a few minor corrections that still need to be addressed. I look forward to receiving your revised version soon so that the manuscript can proceed toward acceptance.

Best regards,
Armando Sunny

Reviewer 1 ·

Basic reporting

Review of manuscript #104236

General comment

This study reports the findings from a rapid assessment survey of benthic species on natural and artificial substrata carried out around Guanabara Bay, a bay that is likely associated with the introduction of non-native species due to high propagule pressure. This survey revealed the continued spread of the non-native mussel Perna viridis and the apparent disappearance or range retraction of the non-native sun-coral (Tubastraea sp.) within the bay. SDM of P. veridis along the coast of Brazil showed that some sections of this coast are a suitable habitat for this species, including Guanabara Bay. This study would benefit from a detailed morphological description of how the authors identified benthic species, including the focal non-native species.

Specific comments

Abstract

Line 21 (also line 45): ‘Favoring elements’ is ambiguous, please be more specific.
Line 23-25 (also line 148): It is not specified that the numbers in front of the name of the major groups are the number of taxa.
Line 26: “…, which is considered a highly…”
Line 27 (also line 53): Please add the year to avoid confusion: “Less than six years ago in 20XX”
Line 31: “substrates” should be “substrata”

Introduction
Line 39: “submersed”
Line 45: One might need to explain how chemical pollution, fisheries, etc, facilitate biological invasions instead of just saying ‘favoring elements’. Also, the superlative of “all favoring elements” should be avoided since it is unlikely that all factors can be considered and made known (e.g., escape from predators, etc…)
Line 51: Unclear as to the meaning of “expressive volume”?

Materials and methods
Line 77-84: Perhaps consider placing the geographic coordinates in brackets.
Line 88: Remove the phrase “attest that it cannot be considered a homogenous environment” since that is already understood based on the first part of the sentence and the citation.
Line 98: Can you give some examples of what you categorize as natural and artificial substrata?
Line 110: What resources did you use to taxonomically identify the species?
Line 118, 136, 207, 220: Usually the genus is not abbreviated when it begins the sentence.

Results
Line 146: Would it be possible to prepare a list of all 83 taxa in a supplementary table?
Line 157: “… at Guanabara Bay and other parts of the Brazillian coast”
Line 160: How is recruit and juvenile defined? For instance, is it based on size? Also, Figure 1 is missing scale bars. Was identification based on external characteristics or also included some dissected individuals to ensure that the morphology matches taxonomic descriptions of the species?

Discussion
Line 177: Please remove subjective adjectives such as “exceptional”.
Line 189: “We did not observe any sun-coral…” Is this species non-native to Guanabara Bay? Can you conclude that this species has disappeared locally from the bay? Perhaps you may wish to mention this absence in the abstract too. Is it sp. or spp. (i.e., one species or more)?

Table
Table 1: Please consider an alternative for the use of “N/A” for “native and artificial” because it could be mistaken as “not applicable”.

Figures
Figure 1: Missing scale bars in each of the panels. It would be helpful to name the “other fouling species” and Perna perna and indicate it/them with an arrow in panels C and D.
Figure 2: The dark orange diamond symbol and the red diamond symbol are difficult to tell apart.

Experimental design

A note on how species were identified and what resources were used to identify native and non-native species would be recommended.

Validity of the findings

The findings and data interpretation were appropriate.

Additional comments

No additional comments.

·

Basic reporting

The English is in general very good, but could be improved by consulting a native English speaker.

The background is presented well.
No raw data are included.
There are no hypotheses. The MS simply reports P. viridis at 17 of 19 stations and lists 15 non-native species identified during the survey.

Experimental design

This is OK so far as it goes, but the title refers to Guanabara Bay. The northern and northwestern parts of the bay were not surveyed, but no rational for this is presented.

139 TSS should be indicated as the abbreviation.

Validity of the findings

In recent years the highly invasive Asian green mussel Perna viridis has been reported from a number of localities in the Caribbean Sea and the western Atlantic Ocean. It was first found in Brazil in May 2018 at two sites in Guanabara Bay at Rio de Janeiro, with one individual being 91mm long and fully mature (Messano et al. 2019). Others were at sizes indicating they were reproductively capable. The species has subsequently been reported at other Brazilian localities to the north and south.
This paper reports on a survey undertaken at 19 sites in in Guanabara Bay in September and October 2022, just over four years after the first record in the bay. The first goal of the survey is stated (lines 66 to 68) is to conduct a rapid assessment of sessile species in the bay and to survey non-native species. This conflicts with the title, which is restricted to P. viridis. All information on other non-native species is thus irrelevant to the title. This dichotomy between the title and text occurs throughout the manuscript.
The key information is thus the fact that P. viridis was found at 17 of 19 sites in an estuary where it was already known to be present. No further information such as densities, size frequencies, reproductive condition, interactions with P. perna, etc. is provided. P. viridis has already been reported to the south of Rio de Janeiro. The 15 non-native species recorded were all previously known from the bay and no details of their distribution or abundance are presented; they are simply listed. This manuscript is not suitable for publication in PeerJ and should be rejected.
It is interesting that the congeneric P. perna also introduced to the area and is long established. The two species were found together at some sites, but not others. It would be interesting to undertake a study examining competitive interactions between the two species.

Additional comments

Lines 55 and 56. Numbers <10 should be written in full, i.e. two, not 2, and three, not 3.
The paragraph at lines 146-157 states the survey found 83 taxa in a variety of phyla, 15 of which were non-native. All of the non-native species had previously been reported in the bay. This paragraph is irrelevant and should also be deleted from the abstract as the paper is about P. viridis. Table 1 essentially repeats this paragraph.
159 refers to P. viridis at various life stages, but no information is given. Figure 1a is supposedly a recruit and b a juvenile, but no measurements are provided how do you know 1b is a juvenile if P. viridis can mature at 30mm? Figure 1c shows P. viridis co-occurring with other fouling species. Don’t all four of the pictures show this?
Paragraph at 177 should be related to P. viridis.
184-198 is not related to P. viridis.
201-202 is poorly written as P. viridis is native to all of the areas listed.
220-224 implies P. viridis has been dispersing rapidly in Guanabara Bay since they found it at 17 stations and it was previously known from only two. However, the previous study (Messano et al. 2019) only looked at two sites and recorded seven individuals, one of which was 91mm. At this stage one would expect the individual to be fully mature, and others were of sizes where they could spawn. It is likely that P. viridis was already widespread in the bay but was undetected as no surveys had been undertaken.
226 and elsewhere co-generic should be congeneric.
229 neither P. viridis nor P. perna is distributed worldwide.
Figure 2 shows the environmental suitability to be highest in the southernmost part of the map. P. viridis is a tropical species. Assuming sea surface temperatures decrease going south on the Brazilian coast this suggests there is a serious flaw in the model.
The References section is sloppily prepared.
Many genera and species are not in italics: lines 280, 281, 283, 284 …..
The first four reference use three different styles of journal citation: Spelled out in full, with key words in caps (282), abbreviated (287) or in full with only the first word in caps (289). These are only examples; it occurs throughout the section. 305 is a mixture of caps and lower case major words.
Is 297 a proper publication or unpublished communication?
355 and 359 lack a journal citation.
Initials of some authors are followed by a full stop (411) while others are not (414).
da (324) and de (327) are presented differently from De (330). Are these names alphabetically presented correctly or should they be by the surnames? 368 is Messano L de.
The entire manuscript needs to be very carefully edited.

Reviewer 3 ·

Basic reporting

This is an interesting survey about the spreading process of invasive species, and since this point of view represents an improving and needed step to understand how these organisms may impact on the invaded ecosystems. The authors took a step forward, after previous publications in which occasional records of the Perna viridis were reported, doing a more extensive survey in order to assess the increasing distribution of the mentioned species.

Experimental design

One of my main concerns after reading the manuscript was about the method used to quantify and identify the several invasive species present in the estuary. However, after a short research about this method I can see that it is widely used in this type of research. Obviously, I am not familiar with the use of Coral Point Count with Excel Extensions CPCe 4.1, but I guess other lecturers like me would also ask for more details about how this methodology can be identified with enough confidence in the different species. The citation of previous studies in which these species, or similar are identified using this method can for sure help to support the actual manuscript. For example: Aji LP, Maas DL, Capriati A, Ahmad A, de Leeuw C, et al. (2024) Shifts in dominance of benthic communities along a gradient of water temperature and turbidity in tropical coastal ecosystems. PeerJ 12: e17132. The use of Coral Point Count with Excel Extensions CPCe 4.1 has more than 1000 cites, for sure making reference to some of these may help to understand how robust the method is. Especially when the lecturers realized that no extra sampling was performed, at least triplicate samples from some quadrants to use as control of the identification. Or in situ counts like it was done by Maher, R. L., Johnston, M. A., Brandt, M. E., Smith, T. B., & Correa, A. M. S. (2018). Depth and coral cover drive the distribution of a coral macroborer across two reef systems. PLoS ONE, 13(6), e0199462. DOI:10.1371/journal.pone.0199462. It would be good to include some words to justify this gap.
Another comment, but related to the introduction, is that in this section there is a lack of general introduction about the topic of research. It would be good having three or four lines at the beginning describing the problem of introduced or invasive species associated with coastal areas, and the relation with vectors as ballast water and biofouling, as well as about the impact on marine ecosystems. Then the authors can start describing the particular area of research.

Validity of the findings

no comment

Additional comments

Minor comments

Lines 21-25. It is not clear here if the 83 identified taxa are exotic or just the total. Later in the result section it is clear that it is the total amount, but It should be clarified also in the abstract.

Lines 44-47. Why only the introduction of sessile exotic species are emphasized? I would say that these conditions favours the introduction of aquatic exotic species in general. I suggest also modifying the redaction to avoid using so many parentheses, even if that is justified by the rules for species name and authors for first description of those.

Lines 76-84. These data should be on a table or maybe on the legend of the figure 2 rather than in the main text.

Discussion

Please compare your results with other similar studies on this species, as for example:

Adrianto L, Rudianto R, Aliviyanti D, Widodo MS, Amalia M, et al. (2024) Distribution Analysis of Green Mussels (Perna viridis) in Banyuurip Village, Ujung Pangkah District, Gresik Regency. BIO Web of Conferences 92: 01021

Gilg MR, Howard R, Turner R, Middlebrook M, Abdulnour M, et al. (2014) Estimating the dispersal capacity of the introduced green mussel, Perna viridis (Linnaeus, 1758), from field collections and oceanographic modeling. Journal of Experimental Marine Biology and Ecology 461: 233-242.

Line 265-266. Don’t need to use the full name here.

Lines 330. Please check the format of the reference list. Most of the scientific names are not in italic for example.

---

## Round 0.2 · accepted · Accept

Congratulations on the acceptance of your work. We look forward to seeing the impact your research will have on the field and to future submissions from you and your collaborators.

Thank you for choosing PeerJ as the journal to publish your research.

Best regards,

Armando Sunny